# Comparing Performances of Predictive Models of Toxicity after Radiotherapy for Breast Cancer Using Different Machine Learning Approaches

**DOI:** 10.3390/cancers16050934

**Published:** 2024-02-25

**Authors:** Maria Giulia Ubeira-Gabellini, Martina Mori, Gabriele Palazzo, Alessandro Cicchetti, Paola Mangili, Maddalena Pavarini, Tiziana Rancati, Andrei Fodor, Antonella del Vecchio, Nadia Gisella Di Muzio, Claudio Fiorino

**Affiliations:** 1Medical Physics, IRCCS San Raffaele Scientific Institute, 20132 Milan, Italy; ubeira.mariagiulia@hsr.it (M.G.U.-G.);; 2Data Science Unit, Fondazione IRCCS Istituto Nazionale dei Tumori, 20133 Milan, Italy; 3Radiotherapy, IRCCS San Raffaele Scientific Institute, 20132 Milan, Italy; 4Department of Radiotherapy, Vita-Salute San Raffaele University, 20132 Milan, Italy

**Keywords:** AI models, early-stage breast cancer, modeling, radiotherapy, toxicity

## Abstract

**Simple Summary:**

Studies comparing performances of machine learning (ML) methods in building predictive models of toxicity in RT are rare. Thanks to the availability of a large cohort (n = 1314) of breast cancer patients homogeneously treated with tangential fields, different ML approaches could be compared. This work shows how more complex models typically achieve higher performances. At the same time, for this test case, the importance is given mainly by a few variables, and toxicity can be predicted by simpler models with similar performances. The availability of more individually characterizing features (here partially missing) is expected to have a likely much higher impact than the choice of the best-performing ML/DL approach.

**Abstract:**

Purpose. Different ML models were compared to predict toxicity in RT on a large cohort (n = 1314). Methods. The endpoint was RTOG G2/G3 acute toxicity, resulting in 204/1314 patients with the event. The dataset, including 25 clinical, anatomical, and dosimetric features, was split into 984 for training and 330 for internal tests. The dataset was standardized; features with a high *p*-value at univariate LR and with Spearman ρ>0.8 were excluded; synthesized data of the minority were generated to compensate for class imbalance. Twelve ML methods were considered. Model optimization and sequential backward selection were run to choose the best models with a parsimonious feature number. Finally, feature importance was derived for every model. Results. The model’s performance was compared on a training–test dataset over different metrics: the best performance model was LightGBM. Logistic regression with three variables (LR3) selected via bootstrapping showed performances similar to the best-performing models. The AUC of test data is slightly above 0.65 for the best models (highest value: 0.662 with LightGBM). Conclusions. No model performed the best for all metrics: more complex ML models had better performances; however, models with just three features showed performances comparable to the best models using many (n = 13–19) features.

## 1. Introduction

To date, the use of artificial intelligence (AI)/machine learning (ML) models in the medical field is increasing thanks to their ability to learn from training data without being explicitly programmed. For this reason, these techniques are being more and more considered in the field of cancer therapy, going towards an era of “precision oncology” [1,2]. The topics span from medical imaging [3], personalized drug discovery [4], target auto-contouring, and dose distribution calculation [5], to the outcome and toxicity prediction, which is the focus of the current work.

In particular, in the field of breast cancer, the majority of patients [6,7] are proven to reduce local recurrence [8] and overall mortality [6,7,9]. However, breast radiotherapy has several side effects, both acute and late [10,11,12,13]. Early toxicity includes breast erythema and desquamation occurring within 90 days after treatment and generally recovers with time. Late toxicity (e.g., telangiectasia, fibrosis, and hyperpigmentation of the skin) may, on the contrary, often become irreversible and progressive over time.

Developing accurate prediction models for radiotherapy-related side effects (e.g., toxicity) is crucial to minimize their impact [14]. This would allow clinicians to deliver more personalized treatments by identifying the main factors influencing toxicity and incorporating them into the therapy decision and optimization. Although severe late toxicities following adjuvant radiotherapy for breast cancer are relatively rare [6,7,11], the number of patients treated in high-income countries is huge and with an annual increase of about 4% [15], pushing researchers to spend much energy in predicting and, possibly, preventing them. In addition, an association between the intensity of transient acute effects and the insurgence of late toxicity has also been reported [16], keeping high the interest also on predictive models of early toxicities.

ML techniques are widely used for making predictions on new data, but they require a diverse and large enough dataset to be trained on. This is one of the reasons why clinical application is still poor together with a lack of model interpretability [17,18]. Despite this, several attempts to use ML in developing predictive models of toxicity in radiotherapy appeared in the last years [17], including the breast cancer case [19,20,21]. ML holds the potential to enhance prediction capabilities given sufficient patient data. However, studies comparing the performances of ML methods in predicting radiation-induced toxicity, trying to assess the best approach, are largely lacking [17]. In particular, the interpretability issue remains relevant when trying to find the “best” approach: advanced ML/deep learning (DL) methods often result in “black-box” models and/or may tend to include a large set of clinically low-relevance variables, making the picture quite obscure to the clinicians [18]. Metrics typically used to evaluate classifier performance for high-risk toxicity patients, such as the area under the receiver operating system (ROC) called AUC, can be significantly misleading for strong unbalanced data. Thus, metrics like f1 and balanced accuracy should also be considered. Establishing a common preprocessing strategy involving scaling, encoding, and feature cleaning is crucial for comparing model performances.

To our knowledge, few studies comparing different modeling approaches have been conducted on breast toxicity. They are always on limited-size populations and/or consider only a few ML approaches [22,23,24,25]. In particular, these studies were performed on different outcomes (acute toxicity [22,25], acute desquamation [23], radiation-induced dermatitis [24]). The input variables used are diverse, going from spectrophotometric variables [25], baseline characteristics, comorbidities [23], and clinical variables [22], to radiomics features [24]. Some considered small [24,25] or large [22,23] cohorts. Their performances differ depending on the features used. In this regard, our work is better comparable with the one of Rattay et al. [22]. The availability of a large, single-institute cohort of patients, homogeneously treated with whole-breast irradiation using tangential fields with the same fractionation schedule and following similar contouring and planning procedures, made it possible to explore the issue. The rate of events pushed us to focus on early toxicity, considering the number of patients/events as large enough to successfully apply many ML methods. Ref. [25] shows a high performance improvement thanks to the other variables considered.

## 2. Materials and Methods

This study has made use of a large, single-institute cohort (n = 1325) of breast cancer patients homogeneously treated with tangential fields. Patient data were analyzed within the retrospective study approved by the Institutional Ethical committee (registered to ClinicalTrials.gov, Identifier No.: NCT03077191). Concerning the sample details, it is possible to refer to previously published papers [26,27]. A total of 11 patients were excluded, either because they received a boost or because they were treated with the VMAT technique. This study involved a single-institute cohort with the advantage of higher homogeneity, similar patients typology and large numbers, but with the disadvantage of not having an external validation as for multi-institutional studies. All procedures described were implemented in Python version 3.7.9 together with Microsoft SQL Server v15.0 via SQL Server Management Studio version 18.11.1. Preprocessing and modeling analysis were performed using an in-house code (medicalAI in mAItre (Medical Artificial Intelligence Toolkit for REsearch, https://github.com/pymaitre, accessed on 16 January 2024)).

### 2.1. Patient Characteristics, Endpoint Definition, and Available Variables

All considered patients underwent breast conservative surgery for pTis-pT3 pNx-pN1a M0 disease in the period from 02/2009 to 05/2017. Radiotherapy was delivered with whole-breast (WB)-IMRT to a total dose of 40 Gy in 15 fractions, without boost. Most patients showed invasive ductal carcinoma (n = 961, 72.5%), while n = 116 (8.8%) patients showed lobular invasive carcinoma. There were 194 patients with pT stage greater than one (14.6%). Patients also received chemotherapy (n = 371, 28%) and monoclonal antibodies (n = 109, 8.2%). The details of the characteristics of the patients and of the treatment are reported in Table 1, Table 2 and Table 3 and in the papers by Fodor et al. [26,27].

The endpoint was RTOG (Radiation Therapy Oncology Group) G2/G3 acute toxicity, resulting in 204 and 1110 patients with or without the event, respectively. The dataset, including 25 clinical, anatomical, and dosimetric features, was split into 984 for training and 330 for independent internal test. Variables considered for this study were “Age”, “Axillary Dissection”, “Quadrant Position”, “T Size”, “Chemo”, “Type Chemo”, “AB Monoclonal”, “Hormonal Therapy”, “Bilateral RT”, “Right”, “Number of Fractions (fr) with Bolus”, “Bolus”, “PTV Volume”, “PTV V105%” (i.e., volume of PTV receiving more than 105% of prescribed dose), “Body D1%” (i.e., dose received by 1% of the body volume), “Obesity”, “Diabetes”, “Hypertension”, “Thyroid Disorders”, “Smoke”, “Alcohol”, “Asymmetry”, “Overall Cosmesis”, “No Nipple/Retraction”, and “Hormonal Type”. In Table 1, Table 2 and Table 3, the characteristics of the cohort, according to the considered features, are shown, except for a few parameters with low variance and a high LR univariate *p*-value, as explained later.

### 2.2. Data Preprocessing

First, the training dataset was standardized using the robust scaling method [28] (see Table A2 for details), limiting the impact of outliers. The test dataset was consequently scaled accordingly. A one-hot encoder on categorical features (i.e., “quadrant position”; see Table 1, Table 2 and Table 3) was applied, and just the most predictive categories for this specific outcome, out of nine, were selected with a low *p*-value at univariate LR (global < 0.2): “QSE/QSI” and “QSE” i.e., QSE and QSI correspond to the external and internal superior quadrant, respectively). This choice was approved by the referring clinician. Moreover, variables with low intrafeature variance and with a high *p*-value at univariate LR were excluded [29]. Features, indeed, with too-low intrafeature variance (<0.02) cannot be predictive for the model and may just confuse a multivariate model. First, a high *p*-value threshold (global > 0.8) was preliminarily applied to skip features surely not associated with the endpoint. Consequently, “Bilateral RT”, “Alcohol”, “Hormonal Therapy”, “Overall Cosmesis”, “Chemo”, and “AB monoclonal” were thus dropped. Then, Spearman correlation was computed on the remaining 20 features (see Figure 1), and features with Spearman |ρ| > 0.8 were dropped, namely “Bolus”, as it is possible to see in Figure 1. Between correlated features, the one with a lower LR univariate *p*-value was chosen [30]. In Appendix A (Table A4), a summary of how features were selected is reported.

The synthetic minority oversampling technique (SMOTE, see Table A2 for details) was applied to create synthesized data of the minority to compensate for class imbalance. The ratio between minority and majority sample numbers was set to be 0.5. This results in the percentage of classes with/without the event of 32% and 68%, respectively. For completeness, models with a ratio of 1:1 were run, finding better results for some models and worse for others. It was thus decided to keep a more conservative approach, adding fewer synthetic data. Note that SMOTE was not used for the LightGBM and AutoGluon models due to their intrinsic balanced approach.

### 2.3. Statistical and ML Methods

Several ML methods (listed in Table 4 with their acronyms) were considered: LR, Lasso, ElasticNet, KNN, SVM, GNB, MLP, RF, LightGBM, and AutoGluon. Lasso and ElasticNet are part of the same typology of models (i.e., logistic regression). They differ just for the chosen penalty, which is a kind of regularization used to reduce overfitting and to help interpretability. ElasticNet, in particular, is a combination of the other two [31]. KNN is a nonparametric classifier, which uses proximity methods to distinguish between classes [32]. SVM is an algorithm that classifies through the choice of a hyperplane in N-dimensional space [33,34]. GNB is part of the so-called probabilistic classifiers, and it is based on Bayes’ theorem with the assumption of strong independence between input features and of a normal distribution for each class [35]. MLP is a feedforward artificial neural network (ANN) with multiple layers where the mapping between input and output layers is a nonlinear activation function [36]. RF [37] and LightGBM [38] are both algorithms that combine multiple decision trees to reach one final result (decision trees are constructed by a series of nodes in order to split the data), where the second one is typically more accurate than the first one. RF constructs them independently, while LightGBM, which is a gradient-boosting-based algorithm, builds them one after another. This means that data instances with large gradients are kept, while the ones with small gradients are randomly dropped. Finally, AutoGluon is an AutoML code operating in supervised machine learning, typically used for tabular predictions. It is focused on automated stack ensembling, which consists of combining a set of individually trained classifiers to reduce their intrinsic error. Here, it serves as a comparison with the other codes considered [39]. The models previously reported were chosen due to their typical use in ML with regard to classification problems (e.g., [17,40]). AutoGluon and LightGBM are less quoted in previous literature, considering their recent implementation as a future development of previous models.

All models, except for AutoGluon and LightGBM, were run by applying a Bayesian search (see Table A2 for details) to maximize the chosen metric (the best one between balanced accuracy, f1 weighted, f1 macro, and AUC; see ${\mathrm{metric}}_{\mathrm{sfs}}$ in Table 5) on a stratified k-fold cross-validation sample (see Table A2 for details). This means that the training set is split into k = 5 sets of data with the ratio between each patient class fixed. The model is trained using k-1 fold and validated on the one left outside. The results obtained on the k validation sets varying the parameters were then combined by averaging the score defined to choose the hyperparameters that maximize the score for each model, described in Appendix A (Table A1). Moreover, asequential backward floating selection (SBFS, see Table A2 for details) was applied, from which a parsimonious feature number was chosen. This means that the chosen metric (see ${\mathrm{metric}}_{\mathrm{mod}\;\mathrm{opt}}$ in Table 5) must fall inside the minimum between 1% of the maximum metric value and its SD error. The feature number is, indeed, considered as one of the free model parameters optimized, and thus, models possibly end up with a different feature number. For each model, the feature importance was derived together with complementary metric scoring for both training and test datasets. A more complete approach to address explainability would be to use an agnostic global explainer (e.g., SHAP [41] or LIME [42]); however, this was not the main goal of the current work, and it will be addressed in a follow-up paper. For each metric, the whole process was run, and the metric with less discrepancy in AUC between training and internal test was chosen. The metric chosen in ${\mathrm{metric}}_{\mathrm{sfs}}$ is, indeed, itself a free parameter for this study. This is better explained in Appendix A (Table A3).

In addition, LR was applied using two different approaches: the first follows the same process described above, while the second one follows an approach similar to the one described in [43,44,45]. While both the procedures followed the same preprocessing, as described in Section 2.2, a different strategy was used for the feature selector of the second procedure. In particular, a multivariate LR was computed with a fixed number of three or four variables chosen among all the feature combinations selected after computing their correlation. This was performed for each combination by applying bootstrapping (1000 random sampling with replacement) in parallel to select the combination with better performance.

A different approach was followed with regard to AutoGluon and LightGBM. Considering they are provided with data augmentation and feature selection, no SMOTE-augmented data were given to these models. While AutoGluon also provides hyperparameter optimization in its own code, this is not possible with LightGBM, and thus, a more complex tool, called “optuna” [46] (see Table A2 for details), was used. The best parameters chosen for all these models are described in Appendix A (Table A1).

### 2.4. Metrics to Compare ML Methods

Different metrics were used to compare the ML models considered in terms of performance. For the first type of procedure previously described, in addition to the well-known AUC, the metrics used are F1, separately on patients with/without the event; Brier score; and slope/$R^{2}$ of the calibration plot. Balanced accuracy was also used for model optimization and SBFS.

F1 score can be considered as a harmonic mean of the precision and recall metrics. The best value is one, and the worst is zero (Equation (Equation 1)).
(1)$$F1=\frac{2\cdot \mathrm{Precision}\cdot \mathrm{Recall}}{\mathrm{Precision}+\mathrm{Recall}}$$

Precision (or positive predictive value) and recall (or sensitivity) are defined as follows (Equation (Equation 2)):(2)$$\mathrm{Precision}=\frac{\mathrm{TP}}{\mathrm{TP}+\mathrm{FP}},\;\;\;\mathrm{Recall}=\mathrm{Sensitivity}=\frac{\mathrm{TP}}{\mathrm{TP}+\mathrm{FN}}$$

In particular, F1 may be computed for each class separately and then combined in different ways, which give different results if the dataset is unbalanced: micro F1, macro F1, and weighted F1. The first is traditionally called accuracy, the second is the arithmetic mean of F1 classes, and the third is the weighted average of F1 classes considering each class’s numerosity.

The known metric, AUC, is calculated as the area under the curve of sensitivity (Equation (Equation 2)) versus 1-specificity, defined as (Equation (Equation 3)):(3)$$\mathrm{Specificity}=\frac{\mathrm{TN}}{\mathrm{FP}+\mathrm{TN}}$$

From the same two metrics, balanced accuracy can be defined as the average of recall obtained on each class as in Equation (Equation 4):(4)Balanced Accuracy=(specificity+recall)/2

Both F1 and balanced accuracy are typically used in cases with unbalanced classification like this one, with the difference that F1 performs better when the attention has to be focused on positives, while for balanced accuracy, negatives and positives have the same importance.

The Brier score (Equation (Equation 5)) is the mean squared error computed between predicted probabilities ($f_{t}$) and observed values ($o_{t}$), and it is always between 0 and 1. The best model must minimize the Brier score value.
(5)$$BS=\frac{1}{N}\sum_{t=1}^{N}{\left(f_{t}-o_{t}\right)}^{2}$$

For the second procedure applied on LR with bootstrap, instead, a model was considered good when the global and inter variable multivariate LR *p*-value was found below 0.05. Just models with the best bootstrap frequency were kept. Moreover, the remaining ones were ordered to maximize the ones with better AUC average between training and test data, and of them, the best one was taken as the best model.

Finally, in both the procedures, maximizing the Youden J metric (Equation (Equation 6)) was used together with the ROC in order to define the threshold used to discriminate between the two classes.
(6)J=sensitivity+specificity−1

## 3. Results

### 3.1. Models’ Performances

A summary of overall model performances is shown in Table 5 and Table 6 and in Figure 2 where different metrics are compared on training–test dataset on the x-axis. The best settings chosen for the models are described in Table A1. The models are plotted in ascending order, starting from the one with less training–testing discrepancy. LightGBM was the best model in terms of absolute metric value and discrepancy between training and test sets. Good performances are also given by AutoGluon, RF, SVC, and MLP. ElasticNet and LR, more often used in the literature, did not show largely different performances (as also shown in the ROC plots of Figure 3). KNN, instead, was strongly overfitted. Possibly, this is related to the fact that it is a nonparametric classifier, which uses proximity methods to distinguish between classes. The sample is unbalanced, and it is not trivial to define a correct value for hyperparameters (e.g., n neighbors). Moreover, this model deals with groups of data and thus more easily tends to better fit training data (forgetting test) if it does not find strong discrimination groups.

The AUC of the test data is slightly above 0.6 overall. The f1 score on patients with toxicity is around 0.3, while it depends on the model for patients without toxicity (mostly above 0.7). A calibration plot was derived for every model, from which the slope, $R^{2}$, and Brier score were extracted. AutoGluon and LightGBM showed the best Brier score, while MLP, RF, and GNB showed the worst values in the test set. ElasticNet and LightGBM training and test slopes have similar values, even if ElasticNet has slope values lower than expected, while LightGBM values are higher. This can be due to the probability distribution derived (see Appendix A, Figure A2 for calibration plots). However, considering the similar slope between training and test sets, it could be easily recalibrated. LR, instead, for example, even if its training slope is near to one, is more difficult to analyze considering the large discrepancy with the test set slope, which could create difficulties in re-calibration. RF slopes were very discrepant if we consider the training data augmented. However, the training slope becomes similar to the one of the test set (see Appendix A, Figure A1) if the same model is applied considering the original training data. MLP resulted in the worst difference between training and test slopes, both considering the original training set and the SMOTE-augmented one. Finally, regarding $R^{2}$, RF and LightGBM showed the least discrepant values closer to one, while LR was the most discrepant with a test value closer to zero. The Youden thresholds shown in Table 5 were separately derived for training and test sets from the ROC plots. The values were typically found similar between training and test sets independently from the model used. The absolute values, instead, focus sometimes on the outcome percentage average (∼0.2) or on the value typically used for classification models (∼0.5). In Table 6, a summary of the precision, specificity, sensitivity, f1 score, and AUC of all the mentioned models for both training and test sets is shown. Confusion matrixes were extracted by choosing as discriminant threshold the one derived by the Youden metric from training data, and consequently, the other metrics were derived. For almost all the models, the training data were presented with SMOTE-augmented data, while the test set was considered without it. For AutoGluon and LightGBM, instead, they were augmented inside the code, and thus, the augmentation was not visible here.

Figure 3 shows the ROC for the best models (i.e., ElasticNet, LR, SVC, and LightGBM). Error intervals were derived through a bootstrap of the population considered, and the lines are the resulting mean values. The optimal thresholds shown in Figure 3 were taken from the mean curves, and thus they are slightly different from the ones in Table 6. For the first three models, training data are shown with SMOTE-augmented data, while LightGBM does augmentation inside the code, and it is not visible here. Although the training data were not considered with augmented data, LightGBM was still the one with better test AUC score and less discrepancy between training and test data, less evident than what is shown in Figure 3.

### 3.2. Importance and Number of Selected Features: Measuring Redundancy

In Figure 4, the feature importances obtained for the best models are shown, respectively, for ElasticNet and LR, for which the importance is given by the model coefficient values, and for SVC and LightGBM derived as feature permutation. ElasticNet and LR have similar derived feature importance which, however, are not shared with more performant models such as SVC and LightGBM, where one of the most prominent features is PTV Volume, which has instead a low coefficient value for ElasticNet, and it is not present for LR. Obesity is among all models always in the first four best variables.

Figure 5 shows two examples (ElasticNet and SVC) of the methods used in SBFS, illustrating the effect of limiting feature redundancy. The same plot derived for all the other models is shown in Appendix A (Figure A4). LightGBM and AutoGluon performed a different kind of feature selector, already inside the code, then their plot could not be explicitly derived. Figures show how the performance chosen for models (see ${\mathrm{metric}}_{\mathrm{sfs}}$, Table 5) increased by varying the number of features used. The thick line is computed on a cross-validation sample, and the light error interval is the standard deviation (SD). The maximum in this plot is shown through the vertical line, and it represents the number of features associated with the best performance. To be more conservative, a parsimonious feature number was chosen, such as the chosen metric falling inside the minimum between 1% of the maximum metric value and its SD error (gray vertical line). This criterion was set to be shared between models to make the performance comparison fair. In the plots, the scores obtained in both cases and their confidence interval of 95% are shown. The parsimonious number of features derived is also summarized in Table 5. Importantly, depending on the model, the number of features building the model could be substantially reduced in most cases up to nine variables, without any relevant impact on the model’s performance. Another approach could be to fix a priori the number of variables, but this would not allow quantifying the different convergence of ML models with regard to number of features chosen.

### 3.3. Bootstrap-Based Logistic Regression with Small Number of Features

Finally, as described in Section 2.3, LR models were developed through a previously suggested bootstrap-based method, fixing the number of variables to three or four. ROC results are presented in Figure 6 for both training and test sets. Training data are presented with SMOTE-augmented data. “Axillary Dissection” and “PTV Volume” were shared by the two models. “Quadrant Position_QSE” completed the three-feature model, while “Age” and “Type Chemo” were added to the four-feature model. In the list of ML model performances, the three-variable model gained the third and fourth positions thanks to its high AUC and f1 score, respectively. However, the model with four variables did not confirm all three variables and showed a worse performance in terms of the AUC of the test set compared with most of the models considered. This means that, also, if this method could give very good performances, it is important to know its large instability by changing the number of features fixed.

## 4. Discussion

The issues concerning the application of advanced ML/DL methodologies in training predictive models to predict RT-induced toxicity (and, more in general, outcome after RT) are multiple and complex. As a matter of fact, the growing availability of AI tools and skills makes their use more accessible and affordable, even in this field. Extending the field to include imaging biomarkers, the large number of image-based quantitative features, which may be extracted from medical imaging, largely contributed to the interest toward advanced ML/DL approaches in handling information and training/validating prediction models [47]. A similar process, with likely minor emphasis, is happening even in the case complex dose metrics are extracted from the 3D dose distributions of individual patients, i.e., dosiomics [48,49]. A large variety of AI-based methods may be considered and applied together, considering issues related to the clinical meaning of the resulting models with different attitudes. This may give the impression of generating controversial reactions in the radiation oncology community, ranging from “easy” enthusiasm to deep skepticism. Despite the many open issues, very few studies dealt with the attempt to rigorously compare the performances of models trained with different ML/DL approaches to the same patient’s cohort [17,50]. This is likely due to several reasons, such as the usually limited number of patients/events, the difficulty of setting up different methodologies using common criteria for data processing, and, highly relevant, the possibility of successfully applying “easier” statistic methods as well as a “priori” knowledge (for instance, NTCP models based on DVH reduction). The availability of a reasonably large cohort of patients treated at a single institute in a quite homogeneous way and followed by the consistent collection of toxicity information pushed us to investigate this issue in a real-life clinically relevant scenario as the one of acute toxicity after WBI. We note that the model of logistic regression can be considered as a benchmark model traditionally used and that the AUC metric on the test set can be considered for comparison between models.

With this aim, first of all, a robust preprocessing pipeline shared between all the routines (scaling, encoding, feature cleaning, data augmentation) was created, and 12 different approaches were consistently implemented in the Python environment, aiming to compare the resulting models to highlight strengths and weaknesses of each of them. The setting of the whole machinery needed a lot of effort for proper model tuning to make model performances quantifiable and comparable. As expected, more complex models (e.g., LightGBM and AutoGluon) achieved a higher test score compared to the other models. However, these models typically ended up having more features and thus being more difficult to be clinically interpretable. It was possible to study their importance thanks to feature permutation algorithms, as shown in Figure 4. Anyway, the first three or four features clearly gave the principal contribution. Thanks to methods such as SBFS, it was also possible to see how performances vary with feature number, confirming that the main gain is obtained with the first few features except for KNN, where, in any case, the model is heavily overfitted. More familiar models in the field of predictive models for toxicity, such as LR, gave better performances in test scores when methods with a fixed number of variables and bootstrapping were used. LR run through the first procedure (SBFS starting from all the features) did not find features as predictive as using the LR bootstrapping method. However, these methods seem to be unstable considering the large discrepancy in the performance of the test set of models with three versus four variables, as well as if the training score was similar. Moreover, these models were automatically chosen considering the average between training and test AUC scores, thus introducing a “data leakage” (i.e., information from outside the training dataset is somehow used to choose the best model). This may have caused a too-optimistic result as compared to the case where the cross-validation model was not influenced at all by the internal test, as for more complex models. From the study of a learning curve (Appendix A Figure A3) for both training and test sets, it was also possible to see that the metrics performance is still increasing, suggesting an improvement with more data. Possibly, this may have an impact on ML models, which were not yet able to fully exploit their maximum potential with the available numbers.

Concerning the interpretability of the models, it is without a doubt that the ones with few features have a much larger chance of being preferred for clinical use. In the current case, MLP and LR3 variables would be the best ones representing a good compromise between accuracy and interpretability: LR3 variables are particularly relevant, giving the possibility to easily represent the risk of toxicity in a single plot, taking the “continuous” variable (i.e., the PTV volume) on the x-axis and modifying the logistic curve in case of presence of the two other adverse factors (Axillary node resection and the superior external Quadrant QSE). Of note, the two clinical factors selected by LR3 variables were considered as clinically motivated by the reference clinician after discussion. On the other hand, LR with fixed variables number and bootstrapping, as mentioned, also seems to be quite “unstable”; thus, the current result should not be generalized to other situations. Importantly, future work could address pushing ML models to stay with the same smaller set of features in order to study how they significantly lower (or not) their performance.

The issue of using models too complex with respect to clinical explainability and usability has already been widely addressed (see, for instance, [18]). For all previously discussed issues; thus, a reasonable approach would be to consider more models at the same time to exploit their different strengths, possibly “forcing” models to limit the number of features and finally considering the “best” models based on the compromise between accuracy and interpretability. To this end, it is highly important to include clinicians in the final model evaluation to better assess clinical interpretations and the model’s usability.

The performance of few-feature LR models was similar to the one of more complex ML models, and this is not what one could expect. The relatively low performances are in line with what was previously found with comparable sample size and features selected in the field of breast toxicity after radiotherapy [22], where our results achieve an AUC on the test set of 0.66 and their results report an AUC of 0.65. A reason could be found in the unbalanced dataset with an outcome with not many positives. Dataset more heterogeneously treated [23], shows slightly higher performance, possibly simply reflecting major differences between the sub-groups of patients, compatible with our population that is homogeneously treated and thus without this phenomenon present. Moreover, ML models tend to work better when there is a higher variable number, which contributes to the result. For this reason, they tend to improve their performance with higher variables used. Furthermore, some important variables are missing. In order to obtain better-performing models (say, AUC > 0.80), the availability of other potential predictors would be highly relevant: in the case of toxicity after breast WBI, more accurate DVH information, information on the genotype and/or the phenotype, densitometry characteristics of the breast may all importantly contribute to improving the accuracy of the resulting models, as seldom reported [13,23,24,25]. The availability of more individually characterizing features (here in part missing) is expected to have a likely much higher impact than the choice of the best-performing ML/DL approach. For example, Cilla et al. [25] (with the same outcome: acute toxicity) reached performances of AUC = 0.87 for LR model using a small dataset (n = 129) and spectrophotometric features, not easy to extract for each patient and not present in our database. On the other hand, the availability of highly detailed individual information is harder to obtain in large cohorts like ours. Due to this, regarding the current breast toxicity cohort, ongoing and future studies will focus on the incorporation of skin DVH information and of densitometry/radiomics features in the models, being both individual features that may be recovered. In addition, the study will be extended to late toxicities.

## 5. Conclusions

The comparison between 12 ML models predicting acute toxicity was performed on a large cohort of patients (n = 1314). No model performed the best for all metrics: more complex ML models have better performances; ElasticNet performs better than the typically used LR when run with no fixed number of variables. For all of them, the feature importance was studied to address the explainability issue, finding that typically the main gain is given by the first few features. LR with three fixed features and bootstrap showed performances similar to the ones of more complex models. Thanks to its simplicity and performance, this model feasibly could be tested and possibly implemented into clinical practice. It could be reasonable to suggest, ideally, considering more models at the same time to exploit their different strengths, possibly “limiting” the number of features. The “best” models should be considered based on the compromise between accuracy and interpretability, including clinicians in the final model evaluation, to better assess clinical interpretations and model usability.

## Figures and Tables

**Figure 1 cancers-16-00934-f001:**
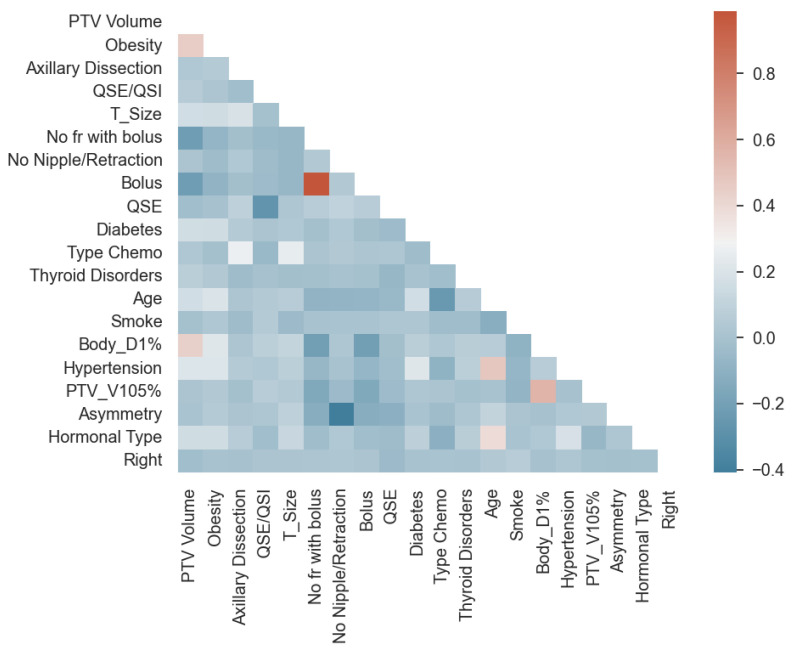
Spearman correlation matrix on the 20 features obtained after preprocessing. Feature order decreases following their LR univariate *p*-value. The color bar shows the correlated Spearman ρ factor.

**Figure 2 cancers-16-00934-f002:**
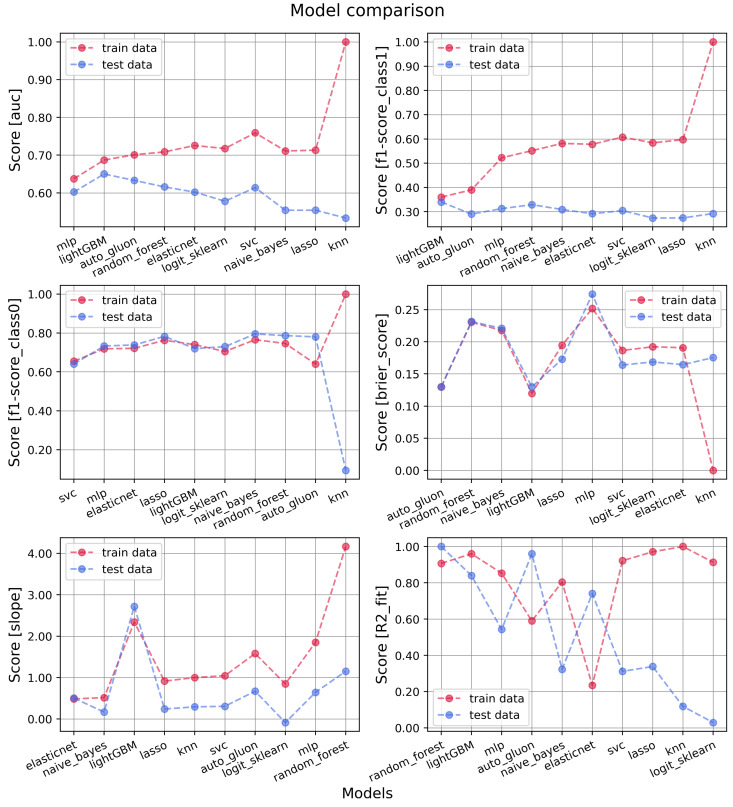
Model comparison on training (red) and test (blue) datasets for acute toxicity. The following metrics were compared: AUC, f1 separately on patients with/without the event, Brier score, slope, and $R^{2}$ of the calibration plot. F1 scores were derived through a threshold chosen by the Youden criterion (see Table 5). Training data of all models, except for AutoGluon and LightGBM, are shown with SMOTE-augmented data, and this is the reason for the large discrepancy.

**Figure 3 cancers-16-00934-f003:**
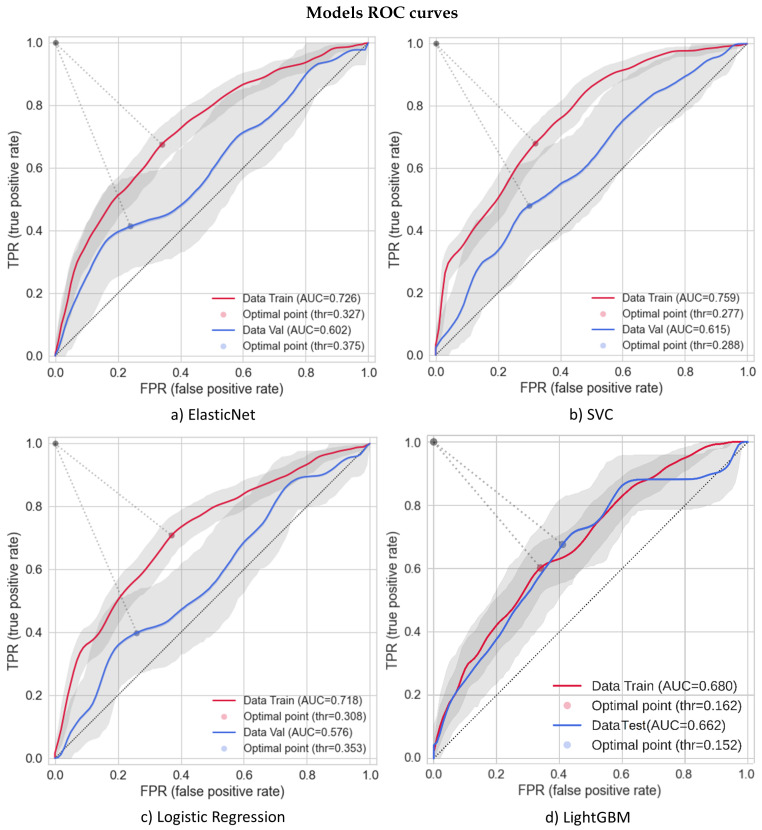
ROC of the acute toxicity outcome with training (red) and test (blue) data. Training data of all models, except for LightGBM, are shown with SMOTE-augmented data. Error bands (gray) and the derived average (lines) are given by a bootstrap computed on training and test sets. The optimal points are the threshold computed through the Youden method on the mean curves.

**Figure 4 cancers-16-00934-f004:**
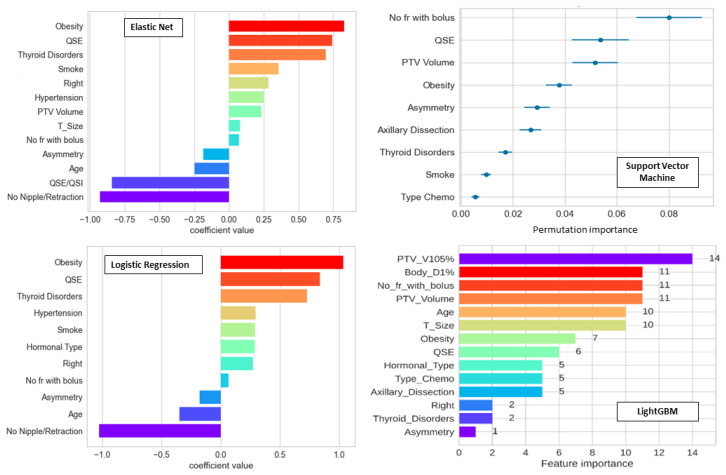
Feature importance of ElasticNet (**top left**), LR (**bottom left**), SVC (**top right**), and LightGBM (**bottom right**) for acute toxicity. For ElasticNet and LR, the values are the coefficients of the model derived on scaled data and thus clinically relevant; for SVC and LightGBM, the values are derived as feature permutation through different intrinsic methods.

**Figure 5 cancers-16-00934-f005:**
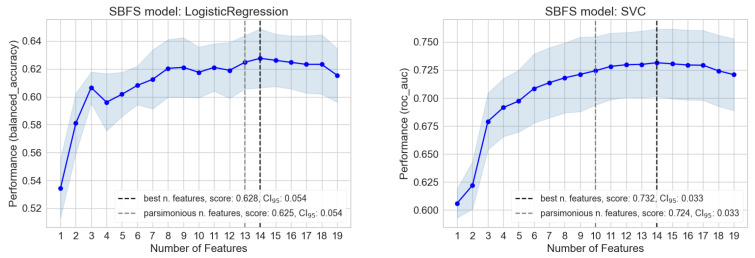
Sequential backward floating selection of ElasticNet (**left**) and on SVC (**right**). The metric performance is computed on the cross-validation dataset. The standard deviation, best, and parsimonious feature numbers are, respectively, represented with a light-blue error band and vertical black and gray lines.

**Figure 6 cancers-16-00934-f006:**
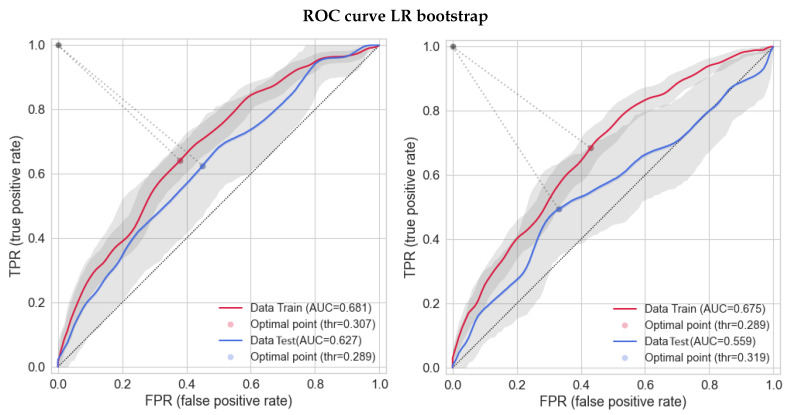
ROC using the method of bootstrap to predict acute toxicity. Training data are shown with SMOTE-augmented data. Models with three (**left**) and four (**right**) features fixed are shown.

**Table 1 cancers-16-00934-t001:** Eleven dichotomic variables selected with intrafeature variance > 0.02 and with *p*-value at univariate LR < 0.8. NA stands for not accorded.

Column Names	No	Yes	NA	%No	%Yes	%NA
Axillary Dissection	1073	241	0	82	17	0
Type Chemo	1022	292	0	78	21	0
Right	677	637	0	52	47	0
Bolus	1007	307	0	77	22	0
Obesity	959	313	42	73	23	3
Diabetes	1119	79	116	85	6	8
Hypertension	751	450	113	57	33	8
Thyroid Disorders	1018	178	118	78	13	8
Smoke	1054	260	0	80	19	0
No Nipple/Retraction	124	1190	0	9	90	0
Hormonal Type	582	732	0	44	54	0

**Table 2 cancers-16-00934-t002:** The categorical variable selected with intrafeature variance > 0.02 and with *p*-value at univariate LR < 0.8. NA stands for not accorded, equivalent to right breast for this table. With * are shown the most predictive categories (univariate LR *p*-value < 0.2).

Quadrant Position	Count	%Count
QSE *	597	45.4
QSI	223	17.0
QIE	142	10.8
QSE/QSI *	100	7.6
QII	96	7.3
Q retroareolar	54	4.1
QIE/QII	47	3.6
QSE/QIE	42	3.2
QSI/QII	12	0.9
NA	1	0.1

**Table 3 cancers-16-00934-t003:** Seven continuous variables selected with intrafeature variance > 0.02 and with *p*-value at univariate LR < 0.8. NA stands for not accorded. Continuous variables where NA values were avoided in deriving the other columns.

Column Names	Median	IQR	Mean	Std	Min	Max	Count NA
Age	62.16	[51.57, 70.65]	61.32	11.92	27.98	91.20	0
T Size	1.30	[0.9, 1.8]	1.43	0.80	0.00	6.70	14
No fr with bolus	0.00	[0.0, 0.0]	1.64	3.07	0.00	15.00	41
PTV Volume	642.25	[445.62, 914.47]	709.51	359.56	114.80	2649.00	0
PTV V105%	2.46	[1.07, 4.39]	3.32	3.45	0.00	33.63	0
Body D1%	41.02	[40.78, 41.27]	41.03	0.42	38.37	46.23	0
Asymmetry	1.00	[1.0, 2.0]	1.22	0.83	0.00	3.00	0

**Table 4 cancers-16-00934-t004:** Models’ description with acronyms and reference paper. Links to the model implementation used are in Table A2. * Model typically used.

Model	Acronym	Reference Paper	Description
Logistic Regression *	LR	[31]	The event probability is a function of a linear combination of independent variables.
Lasso	Lasso	[31]	It is part of the same typology of models of LR, differing for the chosen penalty.
ElasticNet	ElasticNet	[31]	It is a combination of LR and Lasso, regarding the chosen penalty.
k-Nearest Neighbors	KNN	[32]	It is a nonparametric classifier, which uses proximity methods to distinguish between classes.
Support Vector Machines	SVM	[33,34]	It is an algorithm that classifies through the choice of a N-dimensional hyperplane.
Gaussian Naïve Bayes	GNB	[35]	It is part of the so-called probabilistic classifiers, based on the Bayes’ theorem with the assumption of a strong independence between input features and of a normal distribution for each class.
Multi-Layer Perceptron	MLP ^1^	[36]	It is a feedforward artificial neural network (ANN) with multiple layers where the mapping between input and output layers is a nonlinear activation function.
Random Forest	RF	[37]	It combines multiple decision trees, constructed independently, to reach one final result.
Light Gradient Boosting Machine	LightGBM	[38]	Development of RF. It is a gradient-boosting based algorithm, which builds multiple decision trees one after another.
AutoGluon	AutoGluon	[39]	It is an AutoML code, focused on automated stack ensembling of individually trained classifiers to reduce their intrinsic error, here used as a comparison with the other codes.

^1^ Note that the version of MLP here used is a simplified version by sklearn, and thus it does not offer all the potentialities of a more complex one (using TensorFlow), which will be investigated in future work.

**Table 5 cancers-16-00934-t005:** Youden threshold derived for training/test data, number of parsimonious features selected, metrics used for SBFS (${\mathrm{metric}}_{\mathrm{sfs}}$) and model hyperparameters optimization (${\mathrm{metric}}_{\mathrm{mod}\;\mathrm{opt}}$). The value of the training Youden threshold here presented was used for the confusion matrix, while the one of the ROC plots is derived as the average of bootstrap computation (for KNN, the training threshold was derived by adding an ϵ=0.01).

Model	${\mathrm{thr}}_{\mathrm{youden}}$ Train	${\mathrm{thr}}_{\mathrm{youden}}$ Test	n. Features	${\mathrm{metric}}_{\mathrm{sfs}}$	${\mathrm{metric}}_{\mathrm{mod}\;\mathrm{opt}}$
LightGBM	0.162	0.130	14	-	-
AutoGluon	0.177	0.090	19	-	-
LR 3 Variables	0.294	0.283	3	-	-
Random Forest	0.511	0.509	5	AUC	f1 macro
SVC	0.250	0.294	10	AUC	AUC
MLP	0.500	0.489	3	f1 macro	f1 macro
ElasticNet	0.322	0.406	13	balanced acc.	balanced acc.
LR Multivariable	0.303	0.415	11	balanced acc.	balanced acc.
LR 4 Variables	0.264	0.327	4	-	-
Lasso	0.356	0.475	9	balanced acc.	balanced acc.
Naïve Bayes	0.448	0.469	9	f1 weight	f1 weight
KNN	0.010	0.517	10	AUC	AUC

**Table 6 cancers-16-00934-t006:** Model comparison for training and test datasets for the outcome of acute toxicity. Models are in descending order following the AUC performed on test dataset. The confusion matrix was extracted by choosing as discriminant threshold the value derived from the Youden training metric (see Table 6), and from this, the other metrics were derived. f$1_{class0}$ and f$1_{class1}$ are, respectively, the f1 score on classes 0 and 1.

	Train						Internal Test					
Model	Precision	Specificity	Sensitivity	f$1_{\mathrm{class}0}$	f$1_{\mathrm{class}1}$	AUC	Precision	Specificity	Sensitivity	f$1_{\mathrm{class}0}$	f$1_{\mathrm{class}1}$	AUC
LightGBM	0.249	0.662	0.608	0.760	0.350	0.680	0.248	0.673	0.588	0.770	0.350	0.662
AutoGluon	0.249	0.481	0.935	0.640	0.390	0.701	0.216	0.712	0.431	0.780	0.290	0.633
LR 3 variables	0.438	0.583	0.693	0.680	0.540	0.681	0.205	0.568	0.608	0.690	0.310	0.627
Random Forest	0.511	0.714	0.599	0.746	0.551	0.709	0.245	0.709	0.329	0.787	0.329	0.616
SVC	0.472	0.522	0.854	0.655	0.607	0.759	0.206	0.514	0.583	0.640	0.304	0.614
MLP sklearn	0.474	0.678	0.582	0.719	0.522	0.637	0.225	0.637	0.510	0.733	0.312	0.602
ElasticNet	0.497	0.651	0.691	0.721	0.578	0.725	0.219	0.658	0.437	0.738	0.292	0.602
LR sklearn	0.487	0.618	0.728	0.705	0.584	0.717	0.204	0.647	0.417	0.730	0.274	0.578
LR 4 variables	0.417	0.487	0.783	0.610	0.540	0.675	0.169	0.471	0.588	0.610	0.260	0.559
Lasso	0.542	0.720	0.665	0.763	0.597	0.713	0.224	0.733	0.354	0.783	0.274	0.554
Naïve Bayes	0.542	0.735	0.627	0.765	0.582	0.711	0.253	0.747	0.396	0.795	0.309	0.554
KNN	1.000	1.000	1.000	1.000	1.000	1.000	0.173	0.050	0.917	0.290	0.094	0.533

## Data Availability

No data are publicly available due to Ethical Committee constraints.

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
