# Peer review of "Comparing Performances of Predictive Models of Toxicity after Radiotherapy for Breast Cancer Using Different Machine Learning Approaches"

_cancers, 2024, doi:10.3390/cancers16050934_

Round 1

Reviewer 1 Report

Comments and Suggestions for Authors

This study evaluated various machine learning (ML) models for predicting toxicity following breast radiotherapy. I concur with the authors on the scarcity of related works in the field, making this study particularly relevant. I have a few minor suggestions for enhancement:

  1. Introduction: The authors might consider dedicating a paragraph to introducing the application of artificial intelligence (AI) in cancer therapy before delving into ML in breast cancer treatment. References such as Ho (Science. 2020 Feb 28;367(6481):982-3), Siddique et al (Reports of Practical Oncology and Radiotherapy. 2020;25(4):656-66.), Chow (https://doi.org/10.1007/978-3-030-58080-3_143-1) could be incorporated for added context.
  2. Materials and Methods: Given that this is a single-institute study, it's essential to acknowledge the limited scope of data compared to multi-institutional studies. Additionally, could the data be made openly accessible for result reproducibility by other researchers?
  3. Section 2.3: With 12 methods employed in this study, it would be beneficial to include a table listing them and provide justification for their selection. Have other study groups utilized these methods in similar work?
  4. Figure 2: A notable deviation between the train and test data for the knn model is apparent in Figure 2. Could the authors provide insights into this observation?
  5. Discussion: Consider incorporating a benchmark for comparing the performance metrics (e.g., accuracy) of the ML models. This could provide a standard reference point for evaluating the results.
  6. Conclusion: It would be valuable if the authors could offer recommendations regarding the most effective models based on their findings, providing practical guidance to the reader.
Comments on the Quality of English Language

No problem to understand the English.

Reviewer 2 Report

Comments and Suggestions for Authors

I have the following comments:

-Lines 66-68. Please briefly explain (e.g., with 1-2 additional sentences) the main findings from the existing relevant literature (refs 17-20).

Overall, the introduction is a bit too long and should be shortened by at least 20-25%, whereas the open questions on the topic and the detailed background and study purposes should be put more into focus.

-Lines 79-80. Please describe the sample characteristics in more detail - just recalling refs 21,22 is not sufficient.

-In the Discussion section, I think it would be important to systematically compare the study findings with those from the existing literature and to provide an explanation of the performance and potential pros and cons of the predictive models illustrated in the manuscript, also in view of their potential practical application.

Comments on the Quality of English Language

Some minor English editing should be performed.

Round 2

Reviewer 1 Report

Comments and Suggestions for Authors

I am satisfied with the modifications and additional contents from the authors as per my comments. The quality and presentation of this manuscript are improved.

Comments on the Quality of English Language

No problem to understand.

Reviewer 2 Report

Comments and Suggestions for Authors

Thank you for your reply.